# Pharmacological Comparative Characterization of REL-1017 (Esmethadone-HCl) and Other NMDAR Channel Blockers in Human Heterodimeric N-Methyl-D-Aspartate Receptors

**DOI:** 10.3390/ph15080997

**Published:** 2022-08-13

**Authors:** Ezio Bettini, Stephen M. Stahl, Sara De Martin, Andrea Mattarei, Jacopo Sgrignani, Corrado Carignani, Selena Nola, Patrizia Locatelli, Marco Pappagallo, Charles E. Inturrisi, Francesco Bifari, Andrea Cavalli, Andrea Alimonti, Luca Pani, Maurizio Fava, Sergio Traversa, Franco Folli, Paolo L. Manfredi

**Affiliations:** 1In Vitro Pharmacology Department, Aptuit, An Evotec Company, 37135 Verona, Italy; 2Department of Psychiatry, VAMC (SD), University of California, San Diego, CA 92093, USA; 3Neuroscience Education Institute, Carlsbad, CA 92008, USA; 4Department of Pharmaceutical and Pharmacological Sciences, University of Padua, 35122 Padua, Italy; 5Institute for Research in Biomedicine (IRB), Università della Svizzera Italiana (USI), 6500 Bellinzona, Switzerland; 6Department of Anesthesiology, Albert Einstein College of Medicine, Bronx, NY 10461, USA; 7Relmada Therapeutics, Coral Gables, FL 33134, USA; 8Department of Medical Biotechnology and Translational Medicine, University of Milan, 20122 Milan, Italy; 9Swiss Institute of Bioinformatics, 1015 Lausanne, Switzerland; 10Department of Health Sciences and Technology, ETH Zurich, 8092 Zurich, Switzerland; 11Institute of Oncology Research, Southern Switzerland, 6500 Bellinzona, Switzerland; 12The Institute of Oncology Research, Università della Svizzera Italiana, 6500 Bellinzona, Switzerland; 13Veneto Institute of Molecular Medicine, 35129 Padua, Italy; 14Department of Medicine, Zurich University, 8006 Zurich, Switzerland; 15Department of Medicine—DIMED, University of Padua, 35122 Padua, Italy; 16Department of Psychiatry and Behavioral Sciences, School of Medicine, University of Miami, Miami, FL 33136, USA; 17Department of Biomedical, Metabolic and Neural Sciences, University of Modena and Reggio Emilia, 41121 Modena, Italy; 18Department of Psychiatry, Massachusetts General Hospital, Boston, MA 02114, USA; 19Department of Health Sciences, University of Milan, 20122 Milan, Italy

**Keywords:** REL-1017, esmethadone HCL, N-methyl-D-aspartate receptor (NMDAR), dextromethadone, d-methadone, ketamine, memantine, dextromethorphan, MK-801, major depressive disorder (MDD)

## Abstract

Excessive Ca^2+^ currents via N-methyl-D-aspartate receptors (NMDARs) have been implicated in many disorders. Uncompetitive NMDAR channel blockers are an emerging class of drugs in clinical use for major depressive disorder (MDD) and other neuropsychiatric diseases. The pharmacological characterization of uncompetitive NMDAR blockers in clinical use may improve our understanding of NMDAR function in physiology and pathology. REL-1017 (esmethadone-HCl), a novel uncompetitive NMDAR channel blocker in Phase 3 trials for the treatment of MDD, was characterized together with dextromethorphan, memantine, (±)-ketamine, and MK-801 in cell lines over-expressing NMDAR subtypes using fluorometric imaging plate reader (FLIPR), automated patch-clamp, and manual patch-clamp electrophysiology. In the absence of Mg^2+^, NMDAR subtypes NR1-2D were most sensitive to low, sub-μM glutamate concentrations in FLIPR experiments. FLIPR Ca^2+^ determination demonstrated low μM affinity of REL-1017 at NMDARs with minimal subtype preference. In automated and manual patch-clamp electrophysiological experiments, REL-1017 exhibited preference for the NR1-2D NMDAR subtype in the presence of 1 mM Mg^2+^ and 1 μM L-glutamate. Tau off and trapping characteristics were similar for (±)-ketamine and REL-1017. Results of radioligand binding assays in rat cortical neurons correlated with the estimated affinities obtained in FLIPR assays and in automated and manual patch-clamp assays. In silico studies of NMDARs in closed and open conformation indicate that REL-1017 has a higher preference for docking and undocking the open-channel conformation compared to ketamine. In conclusion, the pharmacological characteristics of REL-1017 at NMDARs, including relatively low affinity at the NMDAR, NR1-2D subtype preference in the presence of 1 mM Mg^2+^, tau off and degree of trapping similar to (±)-ketamine, and preferential docking and undocking of the open NMDAR, could all be important variables for understanding the rapid-onset antidepressant effects of REL-1017 without psychotomimetic side effects.

## 1. Introduction

N-methyl-D-aspartate receptors (NMDARs) are ionotropic receptors gated by two ligands, glycine (or D-serine) and glutamate, and by the electric potential of the cell membrane. At resting membrane potential, Mg^2+^ is situated in the channel pore formed by the four NMDAR subunits. When NMDAR channels are coincidentally in the open structural conformation and free from Mg^2+^, they allow tightly regulated, subtype-specific Ca^2+^ influx. The co-occurrence of open conformation and Mg^2+^ release can be elicited by excitatory synaptic transmission with presynaptic glutamate release. When glutamate binds to NMDARs, changing their conformation from closed to open, alpha-amino-3-hydroxy-5-methyl-4-isoxazole propionic acid receptors (AMPARs), also activated by glutamate, determine AMPAR-mediated membrane depolarization and Mg^2+^ release from the NMDAR channel pore, allowing Ca^2+^ influx. This phasic, time-controlled, subtype-specific Ca^2+^ signaling via NMDARs determines downstream effects that modulate synapses, forming the basis of structural and functional memory [1,2]. Tonic NMDAR-mediated Ca^2+^ currents at resting membrane potential also require the coincident opening of the NMDAR structural conformation and release of Mg^2+^ from the channel pore. The physiological role of tonic NMDAR-mediated Ca^2+^ currents is increasingly recognized [3,4]. Excessive tonic Ca^2+^ influx mediated by NMDARs may impair the availability of synaptic proteins [3], interfering with stimulus-evoked memory formation induced by phasic NMDAR activation. NMDAR channel blockers reduce excessive tonic Ca^2+^ influx mediated by NMDARs and restore synaptic protein availability, enabling neural plasticity elicited by action potential-mediated, phasic Ca^2+^ influx in animal models of depressive-like behavior [5,6,7,8,9]. NMDAR heterotetramers are composed of two constitutively expressed glycine-binding NR1 subunits, necessary for membrane expression of the functional channel, and two regulatory glutamate-binding NR2 subunits, named 2A, 2B, 2C, and 2D, which differentially regulate Ca^2+^ influx. NMDARs regulate Ca^2+^ influx during phasic and tonic receptor activity. NMDARs containing the 3A and 3B subunits bind only glycine and not glutamate and are not blocked by Mg^2+^ [10], and therefore they were not included in our study.

The uncompetitive NMDAR blockers memantine, esketamine, and the dextromethorphan-quinidine combination are approved by the U.S. Food and Drug Administration (FDA) to treat Alzheimer’s disease, treatment-resistant depression (TRD), and pseudobulbar affect (PBA), respectively. Differences in clinical effects and indications among uncompetitive NMDAR channel blockers appear to be linked to unique receptor-ligand kinetic properties [1,4,11]. Therefore, advances in characterization of the pharmacological properties of therapeutically effective and clinically tolerated uncompetitive NMDAR blockers at the different NMDAR channel subtypes may improve our understanding of NMDAR-related pathophysiology in different disease states, potentially expanding therapeutic options.

REL-1017 (esmethadone HCl; dextromethadone HCl), the d-isomer of racemic methadone, is a low-affinity, low-potency, uncompetitive NMDAR channel blocker, devoid of meaningful opioid agonist effects [12] with experimental evidence for producing brain-derived neurotrophic factor (BDNF) and target of rapamycin complex 1 (TORC1)–dependent antidepressant-like effects [9], similarly to ketamine [5]. In a double-blind, placebo-controlled Phase 2 trial in patients with major depressive disorder (MDD) and inadequate response to standard serotonergic antidepressants, the addition of REL-1017 showed rapid, robust, and sustained antidepressant effects without dissociative side effects [13]. Phase 3 trials of REL-1017 as adjunctive treatment and as monotherapy in patients with MDD are currently ongoing (ClinicalTrials.gov Identifiers: NCT04855747; NCT04688164; NCT05081167). In the context of this work, the current studies aim to characterize the properties of REL-1017 and other uncompetitive NMDAR blockers on cell lines over-expressing specific human heterodimeric NMDAR channel subtypes by fluorometric imaging plate reader (FLIPR) Ca^2+^ assay and by automated and manual patch-clamp electrophysiology. Additionally, we performed [Piperidyl-3,4-(N)]-(N-(1-[2-thienyl]cyclohexyl)3,4-piperidine ([^3^H]TCP) radioligand binding displacement studies with the same molecules in the cerebral cortex of Wistar rats. Finally, we performed in silico studies with REL-1017, esketamine, and arketamine on the four NMDAR 2A-D subtypes, in closed and open conformational states.

## 2. Results

### 2.1. REL1017 Is a Blocker at All NMDAR Subtypes in FLIPR Ca^2+^ Assay

The effect of REL-1017 and other known NMDAR uncompetitive channel blockers (dextromethorphan, (±)-ketamine, memantine, and MK-801) were studied on specific heterodimeric NMDAR subtypes stably over-expressed in Chinese hamster ovary (CHO) cells by measuring intracellular Ca^2+^ levels using a FLIPR assay, performed at saturating L-glutamate concentration and in the absence of Mg^2+^. L-glutamate was selected as the stimulus at 10 μM, a concentration able to elicit maximal Ca^2+^ influx in each cell line expressing heterodimeric NMDAR subtypes NR1-2A, NR1-2B, NR1-2C, and NR1-2D, respectively (Figure 1; Appendix A).

Every test item was applied at 11 different concentrations, ranging from 100 μM to 1.7 nM. Estimated IC_50_ values, Hill slopes, and minimal percentage Ca^2+^ recorded for every test item are reported in Table 1 and Appendix A.

The calculated IC_50_ values of test items were in the expected range: MK-801 was the most potent compound against all NMDAR subtypes, while (±)-ketamine, memantine, and dextromethorphan showed intermediate potencies between MK-801 and REL-1017 (Table 1). MK-801 was more potent against NMDARs containing the 2B subunit, with a measured IC_50_ value of 70 nM. In these experimental conditions (in the absence of Mg^2+^ and at saturating L-glutamate concentration), all compounds except MK-801 demonstrated a greater potency and calculated affinity on the NR1-2C subtype, with IC_50_ values of 23 μM, 3.4 μM, 3.6 μM, and 5.2 μM for REL-1017, (±)-ketamine, memantine, and dextromethorphan, respectively.

Six different concentrations of REL-1017 and other test molecules (50 μM, 12.5 μM, 3.13 μM, 0.781 μM, 0.195 μM, and 0.049 μM) were assayed by FLIPR in the presence of increasing concentrations of L-glutamate to verify the presence of insurmountable NMDAR antagonism, as theoretically expected from uncompetitive channel blockers (Figure 2; Appendix A).

The logistic equation fitting parameters for all tested items are reported in Appendix A. Results with REL-1017, (±)-ketamine, memantine, dextromethorphan, and MK-801 confirmed an insurmountable profile in a FLIPR Ca^2+^ assay, as expected for uncompetitive blockers. The data for all compounds were fitted to an operational equation for allosteric modulators [14,15,16] to estimate K_B_ and % affinity ratio for each test item (Table 2; Appendix A).

Estimated K_B_ values for five NMDAR channel blockers were obtained through FLIPR assay by L-glutamate CRCs, either alone or in the presence of six different concentrations of test item. Experiments were conducted for the various test items, as exemplified in Figure 2 for REL-1017. Operational equation for allosteric modulators was used to estimate K_B_ for all tested molecules, using the formula described in the Section 4.

The estimated equilibrium dissociation constant, K_B_, was measured to be in the μM range for all tested molecules, except MK-801, which demonstrated a sub-μM K_B_. MK-801 was able to reduce the effect of L-glutamate by more than 50% in all NMDAR subtypes beginning at 781 nM. The estimated K_B_ for MK-801 was ≤150 nM in all NMDAR subtypes. The % affinity ratio was computed from estimated affinities, which are the reciprocal of K_B_ values, arbitrarily setting the highest affinity for each NMDAR subtype as 100%. From this analysis, MK-801 showed a preference (100% affinity ratio) for the NR1-2B subtype, while all remaining test items showed a preference for the NR1-2C subtype (Table 2). REL-1017 K_B_ values were measured to be 8.9 μM, 6.1 μM, 4.5 μM, and 7.8 μM for NMDAR subtypes 2A, 2B, 2C, and 2D, respectively. REL-1017 K_B_ values were higher than (±)-ketamine K_B_ values with affinity ratios of 2.1, 5.5, 9.8, and 5.6 for 2A-, 2B-, 2C-, and 2D-containing receptors, respectively. Notably, the L-glutamate concentration-response curve (CRC) in this FLIPR assay showed preferential activation of the NR1-2D subtype than at lower L-glutamate concentrations (Appendix A).

### 2.2. REL-1017 Shows a Preference for NR1-2D Subtype Composition in the Presence of Extracellular Mg^2+^

Under physiological conditions, the presence of extracellular Mg^2+^ blocks the NMDAR channel pore at resting membrane potentials [1]. Since in FLIPR assays membrane potential cannot be controlled, NMDAR-dependent intracellular Ca^2+^ changes cannot be properly measured at physiological Mg^2+^ concentrations (1 mM), where most NMDAR channels at resting membrane potentials would be blocked. Hence, to better characterize the potential activity of REL-1017 in vivo, we examined its effect in automated whole-cell patch-clamp experiments in the presence of physiological concentrations of extracellular Mg^2+^ (Figure 3; Appendix A).

We first verified that 1 mM MgCl_2_ was able to elicit a voltage-dependent blockade when NMDARs were activated by 10 μM L-glutamate and 10 μM glycine (Appendix A) and showed that voltage-dependent Mg^2+^ block is more robust in NR1-2A and NR1-2B subtypes as compared to NR1-2C and NR1-2D, confirming results previously obtained by Kuner and Schoepfer [17]. The current recorded at −60 mV was normalized to a −30 mV current value, which was arbitrarily set to be the 100% value for each NMDAR subtype. The −60 mV current was only 24 ± 2.4% and 26 ± 1.8% (mean ± standard error of the mean [SEM], *n* = 4) for NR1-2A and NR1-2B subtypes, respectively, while it was 64 ± 5.6% and 55 ± 1.8% (mean ± SEM, *n* = 4) for NR1-2C and NR1-2D subtypes, respectively. 10 μM REL-1017 significantly reduced both 10 μM and 1 μM L-glutamate–elicited currents at all measured voltages between −30 mV and −80 mV with the NR1-2D subtype. 10 μM REL-1017 also significantly reduced 10 μM L-glutamate–elicited currents at all measured voltages between −30 mV and −80 mV in NR1-2A subtypes. However, 10 μM REL-1017 was inactive on 1 μM L-glutamate–activated NR1-2A, NR1-2B, and NR1-2C subtypes and on 10 μM L-glutamate–activated NR1-2C subtypes. When NR1-2B was activated by 10 μM L-glutamate, 10 μM REL-1017 significantly reduced the elicited current only at −30 mV, −40 mV, and −50 mV.

To evaluate REL-1017 activity on specific NMDAR subtypes in the presence of extracellular Mg^2+^ (1 mM MgCl_2_), a CRC was performed in a manual patch-clamp assay on NMDARs activated by sub-saturating 1 μM L-glutamate at a voltage of −60 mV (Table 3; Appendix A).

IC_50_ values for REL-1017 were measured to be 63.1 μM, 41.7 μM, 28.4 μM, and 13.5 μM for NR1-2A, NR1-2B, NR1-2C, and NR1-2D subtypes, respectively, confirming the preference of REL-1017 in blocking NMDARs of the NR1-2D subunit composition at physiological Mg^2+^ concentrations, as observed by automated patch-clamp using a single REL-1017 concentration (10 μM).

### 2.3. REL-1017 and (±)-Ketamine Show Similar Degree of Trapping in NMDARs

We compared REL-1017 and (±)-ketamine onset and offset kinetics, as well as “trapping” properties, as these parameters might be relevant for their differential pharmacological effects [18]. With the exception of MK-801, the tested NMDAR blockers in clinical use (ketamine, memantine, and dextromethorphan) or with potential for clinical uses (REL-1017) all showed the lowest IC_50_ and K_B_ values in recombinant cells expressing the NR1-2C subtype in the FLIPR assay. Therefore, we selected the NR1-2C subtype for this study. Future trapping studies with the NR1-2D subtype may add to the NR1-2D data. A whole-cell manual patch-clamp recording assay demonstrated that 10 μM REL-1017 resulted in a 75% reduction of NR1-2C–mediated current, while (±)-ketamine at 10 μM, 3 μM, 1 μM, and 0.3 μM produced 97%, 90%, 75%, and 44% inhibition, respectively, in the absence of extracellular Mg^2+^ (Figure 4; Appendix A).

Therefore, kinetic parameters of the two items were evaluated (Figure 5; Table 4; Table 5) at concentrations eliciting a similar effect (75% inhibition), i.e., 10 μM and 1 μM for REL-1017 and (±)-ketamine, respectively.

10 μM REL-1017 onset and offset kinetic parameters (tau on and tau off) were measured at 4.7 ± 0.21 s and 17.7 ± 1.0 s (mean ± SEM, *n* = 11), respectively. 1 μM (±)-ketamine tau on and tau off were similar at 4.7 ± 0.14 s and 15.2 ± 0.63 s (mean ± SEM, *n* = 10) each. 10 μM (±)-ketamine tau on and tau off were 0.99 ± 0.05 s and 17.2 ± 3.0 s (mean ± SEM, *n* = 4), respectively, with a significantly lower tau on value as compared to 1 μM (±)-ketamine. This was expected, as tau on kinetics are concentration dependent. When comparing equal concentrations at 10 μM (±)-ketamine and 10 μM REL-1017, the (±)-ketamine tau on value was significantly different from the tau on of REL-1017, indicating an approximately 10-fold higher potency, consistent with the FLIPR K_B_ results at the same NR1-2C NMDAR subtype.

We employed a previously described electrophysiological protocol [18] to measure the degree of each compound “trapping” in NMDARs for 10 μM REL-1017 and 1 μM (±)-ketamine (Figure 6; Table 6).

The blockade caused by 10 μM REL-1017 and 1 μM (±)-ketamine was significantly different, with values of 84 ± 1% (mean ± SEM, *n* = 13) and 74 ± 1% (mean ± SEM, *n* = 11), respectively. However, after 120 s of washout without agonist present, there were no significant differences in trapping blockade between REL-1017 and (±)-ketamine, with values of 85.9 ± 1.9% (mean ± SEM, *n* = 13) and 86.7 ± 1.8% (mean ± SEM, *n* = 11), respectively. These findings are consistent with the measured off-rate kinetics, which were similar for the two compounds and are consistent with the known correlation between off-rate and degree of trapping.

### 2.4. Glutamate Receptor Radioactive Binding Assays in Rat Cortical Membranes

To evaluate the binding of MK-801, (±)-ketamine, memantine, dextromethorphan, and REL-1017 to the NMDAR, displacement binding assays were performed in rat cortical membranes using the [^3^H]TCP radioligand. The IC_50_ and K_i_ were measured to be 3.52–2.39 nM for MK-801, 0.55–0.37 μM for (±)-ketamine, 0.79–0.53 μM for memantine, 2.06–1.40 μM for dextromethorphan, and 6.21–4.21 μM for REL-1017. We also tested binding affinities for all test items at a 10 μM concentration at other NMDAR sites (glutamate, glycine, and polyamine), other ionotropic glutamate receptors (AMPARs and kainate receptors), and at metabotropic glutamate receptors. Results at these additional receptors at a 10 μM concentration were below the 50% cutoff for all tested compounds, confirming below threshold affinity for other NMDAR binding sites and other glutamate receptors.

### 2.5. Computational Modeling

Chou and colleagues [19] recently deposited the structure of NMDAR in its open and closed conformation in the Protein Data Bank (PDB). To understand the structure-activity relationship (SAR) of REL-1017, arketamine, and esketamine and their binding mode inside the channel pore, we performed computational docking calculations for the open and closed receptor conformations of each receptor subtype (Appendix A). Figure 7 shows the structure of the NR1-2D receptor (panel A) and the positioning of REL-1017, arketamine, and esketamine at the NR1-2D channel pore in the open (panels B, C, D) and closed (panel E, F, G) conformations.

Our calculations demonstrate that REL-1017 has a higher preference for binding the open-channel conformation compared to arketamine and esketamine, as shown by their difference in docking scores, confirmed at the four receptor subtypes (Table 7). Glide docking score was parametrized to reproduce binding free energies; therefore, a more negative value corresponds to a higher affinity for the protein target.

The computational analyses shown in Figure 7 indicate that esketamine and arketamine bind the closed conformation of the receptor in a similar position, which is situated above the selectivity filter (delimited by the residue Asn615 for the four subunits). The binding of both REL-1017 and esketamine in the open conformation was deeper in the channel compared to the binding position observed for the closed conformation and was deeper for esketamine compared to REL-1017 (Figure 7; Appendix A). Overall, these data appear to support the hypothesis that REL-1017 may have a higher preference for docking the NMDAR channel in the open conformation and a higher propensity for undocking from the NMDAR channel in the open conformation when compared to arketamine and esketamine.

## 3. Discussion

NMDAR channel pores of NR1-2A-D subunit compositions are gated into their open configuration upon the binding of two molecules of glycine or D-serine to the agonist domain on each NR1 subunit and the concurrent binding of two L-glutamate molecules to the agonist domain on each NR2 subunit. During phasic channel activation, presynaptic glutamate release induces AMPAR-mediated membrane depolarization leading to the release of Mg^2+^ from the NMDAR channel pore, initiating time-controlled, subtype-specific influx of Ca^2+^. Stimulus-evoked, action potential-mediated Ca^2+^ signaling through NMDARs directs cellular signaling cascades that modulate synapses, forming memory [1,2]. The physiological role of tonic Ca^2+^ signaling has been related to local availability of synaptic proteins and neuronal maturation [1,3,4,6,7,8]. Well-regulated tonic Ca^2+^ currents, which are graded by “spontaneous” NMDAR-mediated miniature excitatory postsynaptic currents (NMDAR-mEPSCs), may be essential to ensure availability of synaptic proteins necessary for neural plasticity elicited by stimulus-evoked postsynaptic currents (NMDAR-eEPSCs) [3,6,7,8]. Excessive tonic NMDAR-mediated Ca^2+^ currents can be downregulated by uncompetitive NMDAR antagonists, such as ketamine and esmethadone, restoring neural plasticity and resolving depressive-like behavior in animal models [5,6,7,8,9] and in patients [13,20]. The uncompetitive NMDAR channel blocker dextromethorphan has also shown efficacy without psychotomimetic effects in a Phase 2 trial of patients with MDD [21].

In this study, we performed in vitro and in silico characterizations of REL-1017 and other clinically relevant uncompetitive NMDAR antagonists in CHO cells over-expressing NR1-2A, NR1-2B, NR1-2C, or NR1-2D subtypes, in rat cortical membranes, and in computational models. As demonstrated in FLIPR assays, low glutamate concentrations were most active on NR1-2D subtypes (Appendix A), in agreement with prior studies [22]. As expected, REL-1017 and the other tested uncompetitive antagonists were able to antagonize NMDAR-mediated Ca^2+^ influx triggered by glutamate. In the absence of physiological Mg^2+^, REL-1017, dextromethorphan, memantine, and (±)-ketamine showed some degree of preference for NR1-2C subtypes, as shown by FLIPR assay. However, the subtype preference for REL-1017 was generally low.

In FLIPR assays, REL-1017 showed calculated affinity that was lower compared to the other tested compounds, in alignment with the results of the [^3^H]TCP radioligand binding displacement study in the cerebral cortex of Wistar rats. FLIPR assays were conducted in the absence of Mg^2+^ to allow cytosolic measurements of NMDAR-dependent Ca^2+^ influx; however, Mg^2+^ is physiologically present at the synaptic sites. In the presence of physiological concentrations of Mg^2+^ at resting membrane potentials, persistent tonic Ca^2+^ currents are seen preferentially via NR1-2C and NR1-2D subtypes [1,17] due to their lower affinity for Mg^2+^ compared to NR1-2A and NR1-2B subtypes [23].

In automated and manual patch-clamp electrophysiological recordings, in the presence of physiological Mg^2+^ and low glutamate concentrations, REL-1017 preferentially blocked Ca^2+^ currents mediated by NR1-2D in a voltage-dependent manner. These results are consistent with earlier findings for memantine and (±)-ketamine [24]. Ca^2+^ signaling at resting membrane potential is distinct from stimulus-evoked, AMPAR-mediated, phasic activation [1,4,25]. Tonic Ca^2+^ currents via NR1-2C and NR1-2D subtypes constitute a physiological signaling pathway important for neuronal maturation and plasticity [3,4]. Excess ambient glutamate can cause pathological tonic hyperactivity of NR1-2D subtypes (Appendix A), the subtype preferentially targeted by REL-1017 at physiological Mg^2+^ concentrations.

When experiments are conducted in the presence of Mg^2+^, (±)-ketamine and memantine have also shown similar subtype preference [24]. Furthermore, REL-1017 is positively charged at a physiological pH [26], and therefore it is not surprising that its block of NMDARs is voltage dependent, similar to Mg^2+^, as seen in the automated (Appendix A) and manual patch experiments. The voltage-dependent antagonism exerted by REL-1017 may help to explain the lack of dissociative side effects: when membrane potential shifts toward positive, such as during AMPAR-mediated depolarization, the positively charged REL-1017 molecule is less likely to be docked at its binding site within the channel pore. The voltage-dependent block exerted by REL-1017 does not cause dissociative side effects [13], potentially because it does not interfere with phasic NMDAR-mediated Ca^2+^ signaling at activation membrane potential. Following the completion of a phasic activation, once the membrane repolarizes with negative intracellular potential, Mg^2+^ and REL-1017 may exert their tonic blockade at the channel pore at resting or near-resting membrane potentials (tonic channel blockade), causing their physiological and therapeutic effects, respectively.

NR1-2D subtypes have the highest affinity for glutamate (Appendix A), which makes these subtypes more likely to be in the open configuration due to the effect of ambient glutamate. NR1-2D subtypes have a relatively low affinity for Mg^2+^ and therefore the highest probability of being coincidentally in the open configuration and free from the Mg^2+^ block [17]. NMDARs require glutamate binding for the change into the open conformation and must be concurrently free from Mg^2+^ before allowing Ca^2+^ influx during phasic and tonic activity. NR1-2D subtypes remain open for several seconds rather than a few hundred milliseconds or less of other NMDAR subtypes [1]. This extended Ca^2+^ permeability time after activation may increase the probability for the docking of uncompetitive channel blockers, such as REL-1017, into the NR1-2D subtype channel pore. Therefore, in the presence of physiological Mg^2+^ concentrations, compared to other receptor subtypes, the NR1-2D subtype may be the preferential target for REL-1017 and other clinically tolerated uncompetitive channel blockers.

The IC_50_ values determined by radioligand binding assays in this study (3.52 nM for MK-801, 0.55 μM for (±)-ketamine, 0.79 μM for memantine, 2.06 μM for dextromethorphan, and 6.21 μM for REL-1017) may be informative for determining clinical tolerability, with a lower IC_50_ likely indicative of a higher probability of dissociative effects. However, other characteristics, such as trapping [11], may help explain clinically meaningful pharmacodynamic differences among uncompetitive NMDAR blockers with similar NMDAR affinity. Low trapping is considered the reason for the lack of cognitive side effects of memantine, despite its relatively high NMDAR affinity [11]. This low trapping of memantine may also determine its lack of efficacy in the treatment of MDD [27], despite the relatively high (±)-ketamine-like affinity for NMDARs, which was confirmed in our radioligand displacement assay and FLIPR assay. The degree of trapping of REL-1017 was similar to (±)-ketamine in our experimental conditions. The relatively low NMDAR affinity of REL-1017, compared to the tested uncompetitive NMDAR channel blockers, and its relatively high (±)-ketamine-like trapping, could help to explain the absence of dissociative side effects and the robust, rapid, and sustained antidepressant effects [13]. The lack of neuropathological changes in cortical neurons of rats exposed to high doses of REL-1017 [28], in contrast with other uncompetitive NMDAR antagonists known to produce Olney’s lesions [28,29,30,31,32], may also be related to the relatively lower affinity of REL-1017 at the NMDAR, confirmed in our radioligand binding assays, FLIPR assays, and automated and manual patch assays.

Computational studies of ketamine were recently performed for the NR1-2A and NR1-2B subtypes [33]. In this study, we included the NR1-2C and NR1-2D subtypes in our computational analysis. Compared to NR1-2A and NR1-2B subtypes, NR1-2C and NR1-2D subtypes have a higher probability of being antagonized by uncompetitive NMDAR channel blockers at concentrations potentially therapeutic for MDD and other disorders [24]. Our studies suggest that binding of both REL-1017 and esketamine in the open conformation of NMDARs was deeper in the channel compared to the binding position for the closed conformation. Furthermore, the binding site was deeper in the channel for esketamine compared to REL-1017, suggesting a higher propensity of REL-1017 compared to esketamine to undock from the open conformation. These computational findings, which may also be related to the larger size of REL-1017 compared to esketamine, could also help to explain the lack of dissociative effects of REL-1017, in contrast with esketamine and racemic ketamine. The importance of trapping and voltage dependence for NMDAR blockers has been underscored by Bolshakov and colleagues [34].

Recent data for REL-1017, including clinical data [13,35], experimental preclinical data [9], and the present pharmacological characterization of NMDAR channel blockers, support the hypothesis that excessive Ca^2+^ signaling via tonically and pathologically hyperactive NR1-2D subtypes may in part drive MDD. NR1-2D is the same receptor subtype that is preferentially activated by ambient glutamate and preferentially targeted by REL-1017 and other clinically relevant uncompetitive channel blockers. We therefore hypothesize that, when excessive tonic Ca^2+^ influx is corrected by an uncompetitive channel blocker with NR1-2D preference, such as REL-1017, the resolution of MDD symptomatology [13] may result from normalization of synaptic proteins that allow normal neural plasticity, as has been demonstrated in experimental models of depressive-like behavior using ketamine and REL-1017 [5,6,7,8,9]. Kotermanski and Johnson [24] have demonstrated that uncompetitive NMDAR blockers such as ketamine and memantine may also have a preference for the NR1-2D channel subtype in the presence of physiological Mg^2+^ concentrations. Therefore, the hypothesized mechanism of action of REL-1017 in the treatment of MDD appears to be similar to the mechanisms hypothesized for ketamine in animal models [5,6,7,8,9] and in patients with MDD [36].

Importantly, the approximately 10-fold higher affinity of ketamine at NMDARs, confirmed in the current FLIPR, electrophysiological, and radioligand studies, narrows the safety window of ketamine. Although temporary dissociative effects occur in most patients treated with racemic ketamine and esketamine at doses currently used to treat MDD, the current consensus is that induction of dissociative effects is not necessary for the rapid antidepressant effects of ketamine [37,38]. The low affinity of REL-1017 for the NMDAR spares the block of phasic NMDAR activity, as evidenced by its lack of dissociative effects at therapeutic exposures [13,39]. Despite its low affinity, REL-1017 remains trapped in the NMDAR selectivity pore similarly to ketamine and is therefore able to reduce excessive tonic Ca^2+^ currents, with downstream effects that restore synaptic proteins [40] and exert antidepressant-like effects in animal models [9], increase serum BDNF [35], and rapidly improve MDD in patients [13]. The importance of tonic blockade by uncompetitive NMDARs in determining synaptic protein availability, synaptic enhancement, and resolution of depressive-like behavior in animal models has been anticipated by other authors [5,6,7,8]. The importance of the NR1-2D subtype in the sustained antidepressant actions of arketamine, a less potent NMDAR blocker compared to esketamine, has been suggested by experimental work showing that the sustained antidepressant-like effects of arketamine are lost in NR1-2D subunit knockout mice [41].

The preferential activity for NR1-2D subtypes at resting membrane potential in the presence of physiological Mg^2+^ and the sensitivity of this NMDAR subtype to low-concentration glutamate may explain the downregulating effects of uncompetitive NMDAR channel blockers on tonic Ca^2+^ currents via preferential block of hyperactivated NR1-2D subtypes. In particular, the effects of uncompetitive NMDAR channel blockers at NR1-2D subtypes may be seen when pathologically elevated ambient glutamate induces a higher probability of the open configuration preferentially for the NR1-2D subtype, as shown in this report.

Future studies of REL-1017 blocking effects on NMDAR-mEPSCs in cortical brain slices may add informative data to these results.

## 4. Materials and Methods

### 4.1. Drugs and Reagents

All chemicals were of analytical grade. Dextromethorphan and (±)-ketamine were from Merck Sigma-Aldrich (Milan, Italy). Memantine and (+)-MK-801 were from Bio-Techne/Tocris (Milan, Italy). REL-1017 (dextromethadone HCl) was from SpecGx (Webster Groves, MO, USA).

Cell culture consumables were from Gibco Life Technologies (Milan, Italy), including Dulbecco’s Modified Eagle Medium/F12 (DMEM/F12; Cat # 21331), heat-inactivated fetal bovine serum (FBS; Cat # 10500), penicillin-streptomycin (Cat # 15140), L-glutamine (Cat # 25030), MEM nonessential amino acids (NEAAs; Cat # 11140), blasticidin (Cat # A11139), G418 (Cat # 10131), Zeocin™ (Cat # R250), and TrypLE™ enzyme (Cat # 12604). Hygromycin B (Cat# 10687) and Fluo-4 (F14202) were from Invitrogen (Waltham, MA, USA).

### 4.2. Cell Lines

CHO cell lines stably over-expressing heterodimeric NMDARs composed of human NR1 and 2 subunits were generated by Aptuit, an Evotec company (Verona, Italy). Cell line details are available at ExPASy Cellosaurus database. Research Resource Identifiers (RRIDs) are CVCL_B3TY, CVCL_B3TZ, CVCL_B3U0, and CVCL_B3U1 for CHO-hNR1-h2A, CHO-hNR1-h2B, CHO-hN1-h2C, and CHO-hN1-h2D, respectively. Protein accession numbers were NP_015566, NP_000824, NP_000825, NP_000826, and NP_000827 for NR1, 2A, 2B, 2C, and 2D, respectively. NP_015566 corresponds to the NR1-1a splice variant [42] lacking the N1 cassette but including C1 and C2 cassettes. The four different heterodimeric NMDARs studied were NR1-2A, NR1-2B, NR1-2C, and NR1-2D. Cell lines were cultured for a maximum of 75 passages.

Cells were grown at 37˚C, 5% CO_2_ in DMEM/F12, added with 10% FBS, 1% penicillin-streptomycin, 2 mM L-glutamine, 1% MEM-NEAA, 500 µM ketamine, and selected antibiotics. Selected antibiotics were 10 µg/mL blasticidin, 400 µg/mL G418, 400 µg/mL hygromycin B (except for NR1-2B-CHO), and 300 µg/mL Zeocin™ (for NR1-2B-CHO cells only).

### 4.3. FLIPR Assay

FLIPR assay allowed for indirect measurement of intracellular Ca^2+^ concentration by means of a Ca^2+^-sensitive fluorescent dye (Fluo-4) using the FLIPR instrument (Molecular Devices, San Jose, CA, USA). For these assays, 40 mM stock solutions of the test items were prepared in 100% dimethyl sulfoxide (DMSO), while glutamate and glycine 4 mM stock solutions were prepared in water. 384-well compound plates containing 400 × concentrated test item, glutamate, and glycine were prepared using Labcyte Echo acoustic dispensing system and stored at −20 °C until the day of the experiment. Then, 4 × assay plates were prepared by the addition of 30 µL/well of assay buffer in the presence of 0.5% (v/v) poloxamer 188 solution (Sigma-Aldrich code P5556) on experiment day.

Cells were plated in 384 black/clear bottom plates, at a density of 15.000/well, in the presence of 500 µM ketamine and 10 µg/mL tetracycline to induce receptor expression (except for NR1-2B-CHO cells, which showed NMDAR constitutive expression) and then kept at 30 °C in 5% CO_2_ incubator for 24 h. Plated cells were preloaded for 1 h with 2 μM Fluo-4 Ca^2+^-sensitive dye in presence of 2.5 mM probenecid, an inhibitor of nonspecific anion transport, and 500 μM ketamine to avoid cytotoxicity and then washed with a three-cycle wash in assay buffer. This wash procedure removed all ketamine, as assessed by a consistent full response elicited in positive control wells of every plate by 10 µM glutamate plus 10 µM glycine. Assay buffer composition was 145 mM NaCl, 5 mM KCl, 2 mM CaCl_2_, 1 g/liter D-(+)-glucose, and 20 mM HEPES (pH was adjusted to 7.3 with NaOH). Intracellular Ca^2+^ level was monitored by FLIPR (excitation wavelength at 470–495 nm, emission wavelength at 515–575 nm) for 10 s before and 5 min after the addition of test items. All test items, including L-glutamate (when present) and glycine, were added simultaneously. Glycine 10 μM was always included in every test item addition.

In FLIPR CRC experiments, every test item was assessed at 11 final concentrations: 100 μM, 33 μM, 11 μM, 3.7 μM, 1.2 μM, 412 μM, 137 nM, 46 nM, 15 nM, 5.1 nM, and 1.7 nM. L-glutamate and glycine were both used at a 10 μM final concentration. In FLIPR mode of action experiments, every test item was assessed at six final concentrations: 50 μM, 12.5 μM, 3.13 μM, 0.781 μM, 0.195 μM, and 0.049 μM. L-glutamate was assessed at 11 final concentrations: 100 mM, 1 mM, 100 μM, 10 μM, 3.3 μM, 1.1 μM, 370 nM, 123 nM, 41 nM, 13.7 nM, and 4.6 nM. Glycine was kept at a constant concentration of 10 μM during experiments. Fluorescence was measured as the area under the curve (AUC) during the 5 min immediately after the addition of the test item. FLIPR data were normalized to intracellular Ca^2+^ levels in the presence of 10 μM L-glutamate and 10 μM glycine as 100%, while in the presence of assay buffer as 0% response.

### 4.4. Whole-Cell Automated Patch-Clamp Assay

QPatch HTX (Sophion Bioscience, Denmark) automated planar whole-cell patch-clamp platform was used to assess the effects of REL-1017 on L-glutamate–elicited current on heterodimeric NMDARs in the presence of physiological Mg^2+^. For QPatch experiments, cells were grown in T-flasks first at 37 °C, 5% CO_2_ for 3 to 4 h, then at 30 °C, 5% CO_2_ for 18 to 24 h in DMEM/F12 medium with added 500 μM ketamine and 10 μg/mL tetracycline (except for NR1-2B-CHO cells) without additional antibiotics. Cells were manually detached from the culture dish and automatically dispensed in QPatch wells with extracellular solution in the presence of 1 mM Mg^2+^ and 500 μM ketamine. Extracellular solution composition was 155 mM NaCl, 3 mM KCl, 1.5 mM CaCl_2_, 1 mM MgCl_2_, 10 mM HEPES, and 10 mM D-glucose, adjusted to pH 7.4 with NaOH. Intracellular solution composition was 80 mM CsF, 50 mM CsCl, 0.5 mM CaCl_2_, 10 mM HEPES, and 11 mM EGTA, adjusted to pH 7.25 with CsOH.

Clamped cells were extensively washed in extracellular buffer solution to eliminate endogenous L-glutamate and ketamine, which had been previously added to block excitotoxicity. NMDAR-mediated currents elicited by L-glutamate in the presence of 10 μM glycine and 1 mM MgCl_2_ were recorded at different voltages. Holding potential was −80 mV, and voltage protocol included a depolarizing 2-s step pulse to +60 mV to check the quality of the seal and cell NMDAR expression level, followed by a 2-s ramp-up to holding potential. L-glutamate 10 μM or 1 μM was added during the pulse to +60 mV and washed away at the end of the ramp (Appendix A). L-glutamate addition was repeated in two consecutive voltage protocols, and 10 μM REL-1017 was present during the second L-glutamate addition in the treated group but not in the control group. MgCl_2_ 1 mM was present throughout the L-glutamate addition.

### 4.5. Whole-Cell Manual Patch-Clamp Assay

Cells were grown on poly-D-lysine coated glass coverslips for manual patch-clamp whole-cell recording. Intracellular solution composition was 80 mM CsF, 50 mM CsCl, 0.5 mM CaCl_2_, 10 mM HEPES, and 11 mM EGTA, adjusted to pH 7.25 with CsOH. Extracellular solution composition for on-/off-rate and trapping assays was 155 mM NaCl, 3 mM KCl, 1.5 mM CaCl_2_, 10 mM HEPES, and 10 mM D-glucose, adjusted to pH 7.4 with NaOH. Extracellular solution for REL-1017 CRC also contained 1 mM MgCl_2_. Whole-cell recordings occurred at −70 mV fixed voltage equal to holding potential for onset and offset kinetic and trapping assays using NR1-2C-CHO cells. Whole-cell recordings occurred at −60 mV for REL-1017 CRCs.

The data capture and primary analysis were accomplished using HEKA Elektronik’s PATCHMASTER software (v2.53). Data were filtered at 1 kHz and sampled at 2 kHz. Rapid agonist or agonist-antagonist application was accomplished using a BioLogic RSC-160 perfusion device (BioLogic, Seyssinet-Pariset, France). The system consisted of a custom-made manifold of three tubes, whose tip was placed at around 100 μm from the patch-clamped cell under study.

### 4.6. NMDAR Binding Assay in Rat Brain Cortical Membranes

Multiple concentrations of (+)-MK-801, (±)-ketamine, memantine, dextromethorphan, and REL-1017 were tested in displacement binding assays in rat cortical membranes by Eurofins Panlabs Discovery Services Taiwan (Taipei City, Taiwan). The tritiated [^3^H]TCP ligand was selected for its ability to bind the pore of NMDARs (specific binding 94%, K_d_ 8.40 nM, and B_max_ 0.78 pmol/mg).

Wistar rat cortical membranes were obtained as previously described [43]. A 6.3-mg aliquot was incubated with 4 nM [^3^H]TCP for 45 min at 25 °C. Nonspecific binding was defined by 1 μM dizocilpine ((+)-MK-801). Each tested compound was evaluated at the following concentrations: 1 μM, 3 μM, 10 μM, 30 μM, 100 μM, 3 nM, 10 nM, 30 nM, 100 nM, and 300 nM in 10 mM Tris-HCl, pH 7.4, as an incubation buffer. Membranes were filtered and washed, and the filters were then counted to determine [^3^H]TCP specifically bound, allowing us to calculate IC_50_ and K_i_ for each tested compound.

IC_50_ values were determined by a nonlinear, least squares regression analysis using MathIQ^TM^ (ID Business Solutions Ltd., UK). The K_i_ values were calculated using the Cheng and Prusoff equation [44], using the observed IC_50_ of the tested compound and the concentration of radioligand employed in the assay.

### 4.7. Additional Radioligand Displacement Binding Assays

(±)-Ketamine, memantine, dextromethorphan, and REL-1017 were also submitted to Eurofins Panlabs Discovery Services Taiwan for single-concentration (10 μM) binding assays for the following receptors under the following conditions:

Glutamate, AMPA. Source: Wistar rat cerebral cortex. Ligand: 5.0 nM [^3^H]AMPA. Nonspecific ligand: 1.0 mM L-glutamic acid. Incubation time/temp: 90 min/4 °C. Specific binding: 90%. Incubation buffer: 50 mM Tris-HCl, pH 7.4, 200 mM KSCN.

Glutamate, kainate. Source: Wistar rat brain cortex. Ligand: 6.0 nM [^3^H]kainic acid. Nonspecific ligand: 1.0 mM L-glutamic acid. Incubation time/temp: 2 h/4 °C. Specific binding: 80%. Incubation buffer: 50 mM Tris-HCl, pH 7.4.

Glutamate, metabotropic, mGlu2 human: Source: Human recombinant Chem-1 cells. Ligand: 2.0 nM [^3^H]LY341495. Nonspecific ligand: 5.0 μM LY-354740. Incubation time/temp: 60 min/25 °C. Specific binding: 90%. Incubation buffer: 10 mM KH_2_PO_4_, 100 mM KBr, pH 7.6 (pH adjusted with KOH).

Glutamate, metabotropic, mGlu5 human: Source: Human recombinant CHO-K1 cells. Ligand: 30 nM [^3^H]quisqualic acid. Nonspecific ligand: 1.0 mM L-glutamic acid. Incubation time/temp: 2 h/25 °C. Specific binding: 85%. Incubation buffer: 25 mM HEPES, 1 mM MgCl_2_, 2.5 mM CaCl_2_, pH 7.4.

Glutamate, NMDA, agonism: Source: Wistar rat cerebral cortex. Ligand: 2.0 nM [^3^H]CGP-39653. Nonspecific ligand: 1.0 mM L-glutamic acid. Incubation time/temp: 20 min/4 °C. Specific binding: 90%. Incubation buffer: 50 mM Tris-HCl, pH 7.4.

Glutamate, NMDA, glycine: Source: Wistar rat cerebral cortex. Ligand: 0.33 nM [^3^H]MDL 105,519. Nonspecific ligand: 10.0 μM MDL 105,519. Incubation time/temp: 30 min/4 °C. Specific binding: 85%. Incubation buffer: 50 mM HEPES, pH 7.7.

Glutamate, NMDA, polyamine: Source: Wistar rat cerebral cortex. Ligand: 2.0 nM [^3^H]ifenprodil. Nonspecific ligand: 10.0 μM ifenprodil. Incubation time/temp: 2 h/4 °C. Specific binding: 80%. Incubation buffer: 50 mM Tris-HCl, pH 7.4.

Glycine, strychnine sensitive: Source: Wistar rat spinal cord. Ligand: 10.0 nM [^3^H]strychnine. Nonspecific ligand: 1.0 mM glycine. Incubation time/temp: 10 min/4 °C. Specific binding: 77%. Incubation buffer: 50 mM KH_2_PO_4_, 200 mM NaCl, pH 7.1.

For each assay, quantitation was performed by radioligand binding, and significance was considered when ≥50% of max stimulation or inhibition was attained.

### 4.8. Computational Model of Four NMDAR Subtypes in the Open and Closed Conformational States

Structural models were generated for the four NMDAR subtypes using the homology modeling pipeline and implemented the Schrödinger suite for molecular modeling, leaving all the options to their default values.

Amino acid sequences of the subunits were retrieved from UniProt (Q12879, Q13224, Q14957, and O15399 for 2A, 2B, 2C, and 2D, respectively). The structure of the NR1b-2B subtype from *Rattus norvegicus* recently solved by Chou and colleagues [19] in the open (PDB 6WHT) and closed (PDB 6WHS) conformation was used as a template. Finally, all generated models were optimized using the protein preparation wizard procedure implemented by the Schrödinger suite.

Ligand docking calculation was performed with Glide [45,46], centering the grid on the QRN site [33] (i.e., the asparagine residues that correspond to position 615 in the template structures) and using the standard precision (SP) mode. All ligand structures were prepared by LigPrep software. In the case of ketamine, docking calculations were performed on both enantiomers (arketamine and esketamine).

### 4.9. Operational Equation for K_B_ Estimation

The operational equation for allosteric modulators [14,15,16] was created in GraphPad Prism v8.0 software to estimate K_B_ and α parameters for every test item with the assumption that, as a channel blocker, every test item would be able to produce a compete blockade of agonist response at sufficiently high concentration:

$$Y=E_{MAX}\frac{\frac{\tau \left[A\right]}{EC_{50}\left(\tau +1\right)\;}}{\left(\left(\left(\frac{\left[A\right]}{EC_{50}\left(\tau +1\right)\;}\right)+\left(\frac{\tau \left[A\right]}{EC_{50}\left(\tau +1\right)\;}\right)\right)*\left(1+\frac{\alpha \left[B\right]}{K_{B}}\right)\right)+\frac{\left[B\right]}{K_{B}}+1}$$ where Y is the % effect of L-glutamate, [A] is the L-glutamate molar concentration; E_MAX_ is the maximal possible L-glutamate effect estimated from the four-parameter logistic equation; EC_50_ is the half maximal effective L-glutamate concentration estimated from the four-parameter logistic equation; τ is an arbitrary L-glutamate efficacy value at NMDAR set at τ = 100; [B] is the test item molar concentration; K_B_ is the estimated test item equilibrium dissociation constant; and α is the estimated cooperativity term, which indicates the effect of the test item on the L-glutamate dissociation equilibrium constant for the receptor (i.e., α is the estimated ratio between the L-glutamate equilibrium dissociation constant in the absence and in the presence of the test item, and it is expected to be 0 < α ≤ 1 for a negative allosteric modulator affecting the agonist equilibrium dissociation constant). Calculated α values are reported in Appendix A. Percent affinity ratio was computed from estimated affinities, which are the reciprocal of K_B_, and considering the highest affinity for an NMDAR subtype as 100%.

### 4.10. Onset and Offset Kinetic Calculation

At least *n* = 10 independent cells were analyzed. For each cell, the current in the presence of 10 μM glycine only was set as 0%, while the steady-state current induced after 5 s of 10 μM L-glutamate and 10 μM glycine application was set as 100% (Appendix A). The time constants of onset (tau on, seconds) and offset (tau off, seconds) of test item inhibition of glutamate-induced current were calculated using a first order exponential equation:

First order equation for test item onset: *I*(*t*) = *I*_1_ + (*I*_0_ − *I*_1_) × *e*^−^^*t*/^^*τon*^

First order equation for test item offset: *I*(*t*) = *I*_1_ + (*I*_2_ − *I*_1_) × (1 − *e*^−^^*t*/^^*τoff*^)

I(t) is the current at time t. t is the time (seconds) after test item application or removal in the onset or offset equation, respectively. I_0_ is the current after 5 s of 10 μM L-glutamate and 10 μM glycine application prior to test item application. I_1_ is the current 30 s after application of the test item in the presence of 10 μM L-glutamate and 10 μM glycine. I_2_ is the current 50 s after removal of the test item in the continuous presence of 10 μM L-glutamate and 10 μM glycine. τ_on_ (tau on) is the time constant (seconds) of onset. τ_off_ (tau off) is the time constant (seconds) of offset.

### 4.11. Trapping Calculation

Trapping was calculated as previously described [18]. Blocks of 10/10 μM L-glutamate/glycine–evoked currents were calculated according to the formula *B* = [(*I* − *IB*)/*I*] × 100, where I was determined as the current value derived from a linear extrapolation to the end of the L-glutamate antagonist co-application, and IB was the current measured at the end of L-glutamate/blocker co-application.

The residual block of L-glutamate–evoked currents was calculated according to the formula *B**R* = [(*I*_1__*t*_ − *I*_2__*st*_)/*I*_1__*s**t*_] × 100 (2), where I_1st_ was the maximal current measured 1 s after onset of the first L-glutamate exposure, and I_2nd_ was the maximal current measured 1 s after onset of the delayed second L-glutamate exposure following washout of the blocker from the bath.

The block trapped (BT), or the amount of block remaining at the beginning of the second L-glutamate application as a percent of the initial block produced at the end of the previous L-glutamate/antagonist co-application, was calculated according to the formula

*BT* = *B**R*/*B* × 100 (3), where B and BR were defined as above.

### 4.12. Statistical Analysis

The normality (Shapiro-Wilk) test was performed before any statistical analysis. Only data that passed the normality test (*p* ˃ 0.05) were considered for statistical analyses. Comparisons between groups were performed by Student’s t-test for unpaired data or one-way analysis of variance (ANOVA) followed by the Tukey’s *post hoc* multiple comparison test, when appropriate. In all the experiments, *p* < 0.05 was considered statistically significant.

## 5. Conclusions

The pharmacological characteristics of REL-1017 at the NMDAR, such as relatively low receptor affinity, NR1-2D subtype preference, ketamine-like trapping, and propensity for undocking the NMDAR in the open conformation, may help us understand its actions as a rapid-onset antidepressant devoid of psychotomimetic side effects [13]. Progress in pharmacological characterization of uncompetitive NMDAR channel blockers by in vitro and in silico techniques, in light of experimental [3,5,6,7,8,9] and clinical results [13,21,37] with the same molecules, can improve our understanding NMDAR function in physiology and pathology. Tonically excessive Ca^2+^ currents through NR1-2D NMDAR subtypes expressed by select neuronal populations may be central to the pathophysiology of MDD and potentially other disorders.

## Figures and Tables

**Figure 1 pharmaceuticals-15-00997-f001:**
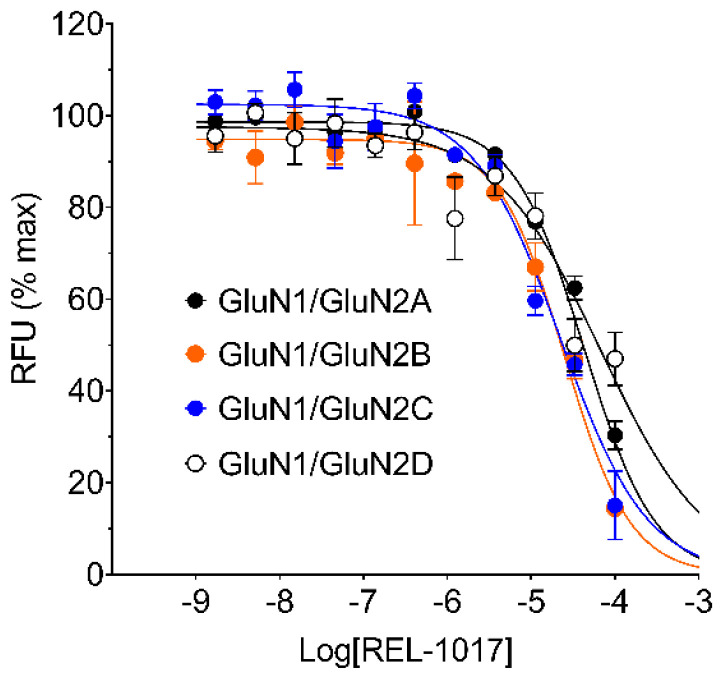
REL-1017 concentration-response curve (CRC) was performed in the presence of 10 µM L-glutamate and 10 µM glycine but in the absence of extracellular Mg^2+^ by fluorometric imaging plate reader (FLIPR) Ca^2+^ assay using Chinese hamster ovary (CHO) cell lines expressing the indicated heterodimeric N-methyl-D-aspartate receptors (NMDARs). REL-1017 caused a decrease in the increased intracellular Ca^2+^ elicited by 10 µM L-glutamate with estimated IC_50_ concentrations of 43 µM, 23 µM, 23 µM, and 68 µM for 2A-, 2B-, 2C-, and 2D-containing NMDARs, respectively. 100 μM REL-1017, the maximal tested concentration, reduced Ca^2+^ influx by 30%, 14%, 15%, and 47% of that evoked by 10 µM L-glutamate for NR1-2A, NR1-2B, NR1-2C, and NR1-2D subtypes, respectively. Fitting parameters for REL-1017 and other known NMDAR blockers, which were tested in identical conditions, are reported in Table 1. Data are mean ± standard error of the mean (SEM), *n* = 6 per group, and were fitted using the four-parameter logistic equation with GraphPad Prism v8.0. RFU is relative fluorescence units.

**Figure 2 pharmaceuticals-15-00997-f002:**
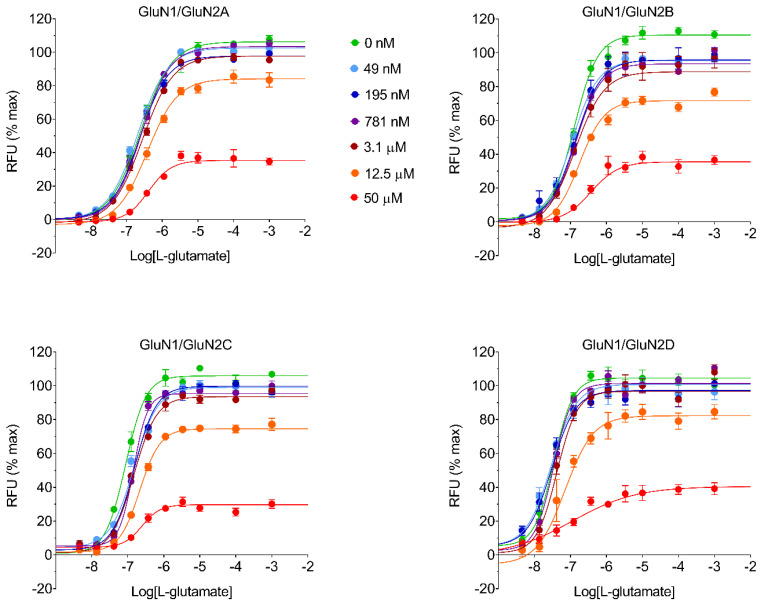
REL-1017 effect on L-glutamate CRC was obtained in a FLIPR Ca^2+^ assay to estimate REL-1017 K_B_. REL-1017 K_B_ is the equilibrium dissociation constant of the REL-1017 receptor complex and the REL-1017 concentration required to occupy 50% of the total NMDAR population. The graphs show L-glutamate CRCs alone (●) or in the presence of six different concentrations of REL-1017 (● 50 µM, ● 12.5 µM, ●3.1 µM, ● 781 nM, ● 195 nM, and ● 49 nM) using four different CHO cell lines expressing heterodimeric human NMDARs: NR1-2A, NR1-2B, NR1-2C, and NR1-2D. An operational equation for allosteric modulators was used to estimate REL-1017 K_B_ and is reported in Table 2. The insurmountable antagonism demonstrated by REL-1017 is apparent from the graphs, as inhibition induced by high concentrations of REL-1017 (e.g., 50 µM) cannot be overcome even by agonist concentrations as high as 1 mM L-glutamate. RFU is relative fluorescence units.

**Figure 3 pharmaceuticals-15-00997-f003:**
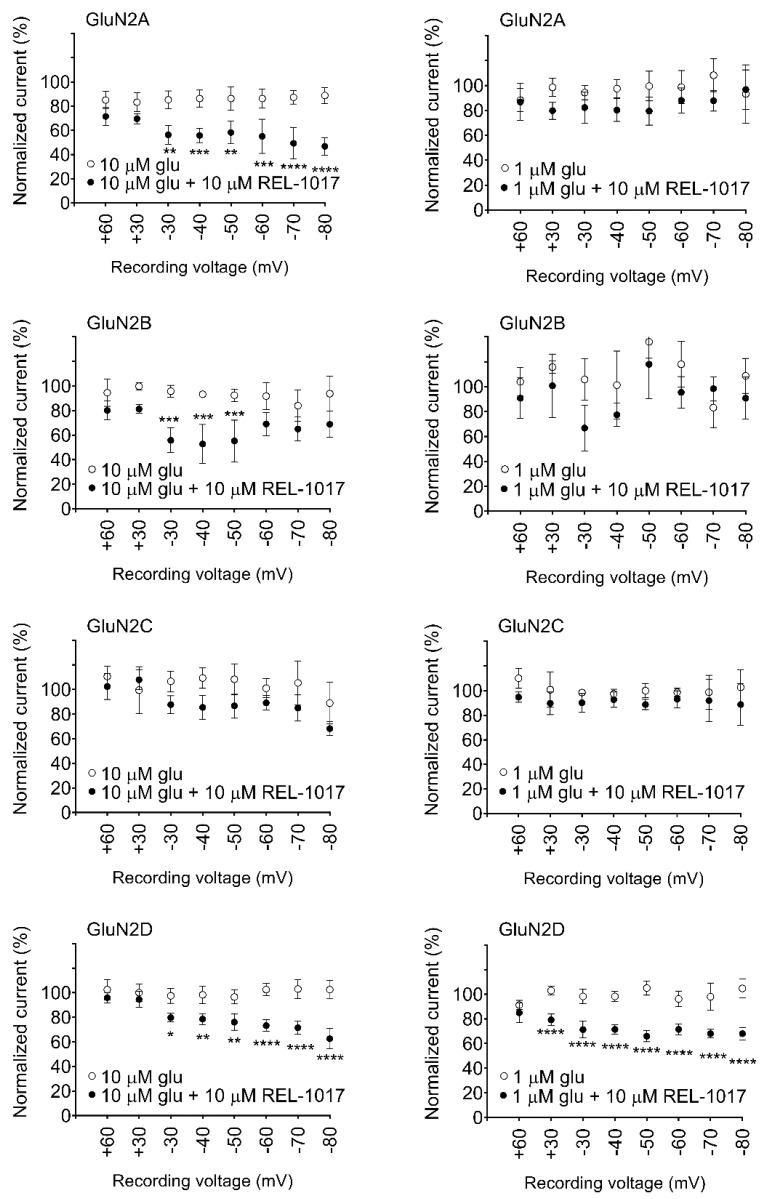
10 µM REL-1017 differentially blocked heterodimeric NMDARs in the presence of extracellular 1 mM Mg^2+^. REL-1017 showed a preference for NMDARs containing the 2D subunit, which is blocked in the presence of either 10 μM or 1 μM L-glutamate at −30 mV or more negative voltages. Data were obtained in automated patch-clamp platform. Every data point represents mean ± SEM (*n* = 4) of normalized current elicited by 10 mM L-glutamate at indicated voltage, measured during a hyperpolarizing ramp, in the absence (○) or presence (●) of 10 µM REL-1017. Statistical results of one-way analysis of variance (ANOVA) followed by Tukey’s multiple comparisons test also reported *p* < 0.05 (*), *p* < 0.01 (**), *p* < 0.001 (***), and *p* < 0.0001 (****). Glu is glutamate.

**Figure 4 pharmaceuticals-15-00997-f004:**
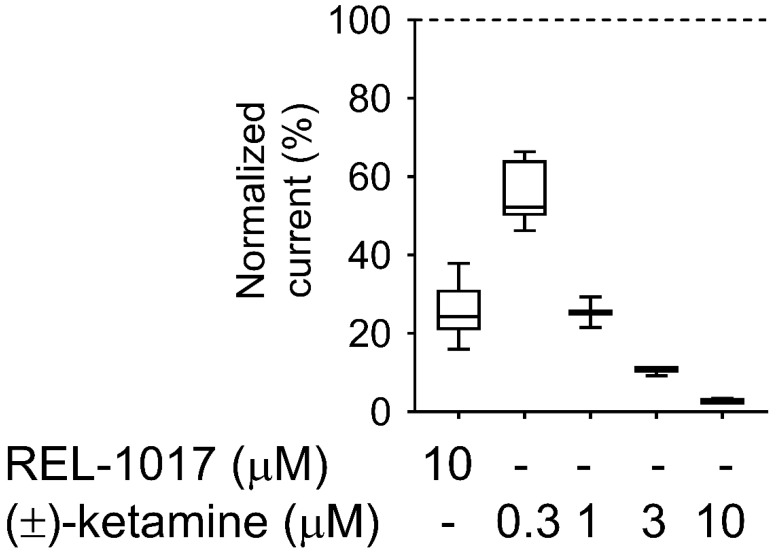
Experiments were conducted using whole-cell patch-clamp electrophysiology at a −70 mV holding potential. Concentrations of (±)-ketamine able to inhibit 75% of the current elicited by 10 µM glutamate and glycine each were applied to a cell line expressing heterodimeric NR1-2C NMDARs to carry out kinetic studies. These experiments showed that 10 µM REL-1017 and 1 µM (±)-ketamine elicited similar current blockade of approximately 75%. Control current (100%) was induced by 10 µM L-glutamate and 10 µM glycine, respectively, and resulted in −594.2 ± 103.7 pA (mean ± SEM, *n* = 28). 10 µM REL-1017 and 10 µM, 3 µM, 1 µM, and 0.3 µM (±)-ketamine reduced L-glutamate/glycine–elicited current by 74.6 ± 1.9% (*n* = 12), 97.2 ± 0.3% (*n* = 3), 89.7 ± 0.6% (*n* = 3), 74.6 ± 2.2% (*n* = 3), and 44.2 ± 3.0% (*n* = 7), respectively.

**Figure 5 pharmaceuticals-15-00997-f005:**
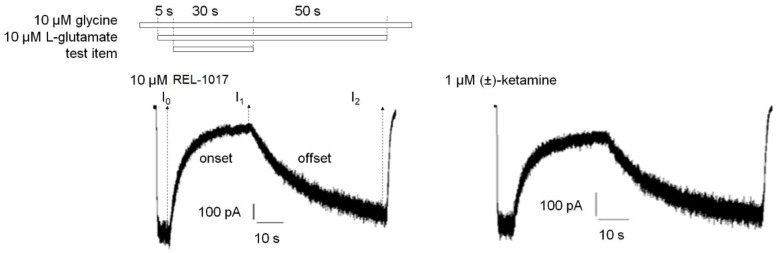
Sample traces of 10 µM REL-1017 and 1 µM (±)-ketamine onset and offset kinetics. Experiments were conducted by whole-cell patch-clamp electrophysiology at a −70 mV holding potential in NR1-2C–expressing CHO cells. A protocol was established to measure how fast REL-1017 blockade of NMDAR-mediated currents can be established (onset kinetic) and how fast this blockade can be removed (offset kinetic) by perfusion with a buffer containing the agonist L-glutamate but devoid of REL-1017. Test item application protocol diagram (**top**) and sample traces (**bottom**) of test item onset and offset kinetic experiments are shown with 10 µM REL-1017-treated cell (**left**) or 1 µM (±)-ketamine–treated cell (**right**). I_0_, I_1,_ and I_2_ were the currents measured at the end of the first 5 s of 10 μM/10 μM L-glutamate/glycine application, the 30-s co-application with test item, and the final 50-s co-agonists application, respectively. In this case, s is seconds.

**Figure 6 pharmaceuticals-15-00997-f006:**
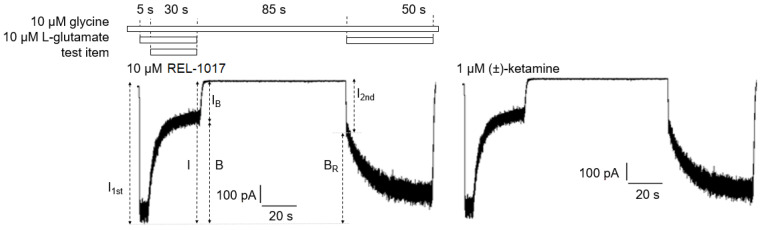
Trapping experiments were conducted in whole-cell patch-clamp electrophysiology at −70 mV holding potential. Trapping experiments were designed to measure trapped block (B_T_) of 10 µM REL-1017 or 1 µM (±)-ketamine that is % ratio between residual block (B_R_) after extensive (85 s) cell wash with glycine alone and initial test item block (B). Test item application protocol diagram (top) and sample traces (bottom) of trapping experiments with 10 µM REL-1017-treated cell (left) or 1 µM (±)-ketamine-treated cell (right) are shown. *I*_1st_ and *I*_2nd_ were the peak currents measured within 1 s after onset of the first and second 10 µM/10µM L-glutamate/glycine addition, respectively. Results from trapping experiments are reported in Table 6. s is seconds.

**Figure 7 pharmaceuticals-15-00997-f007:**
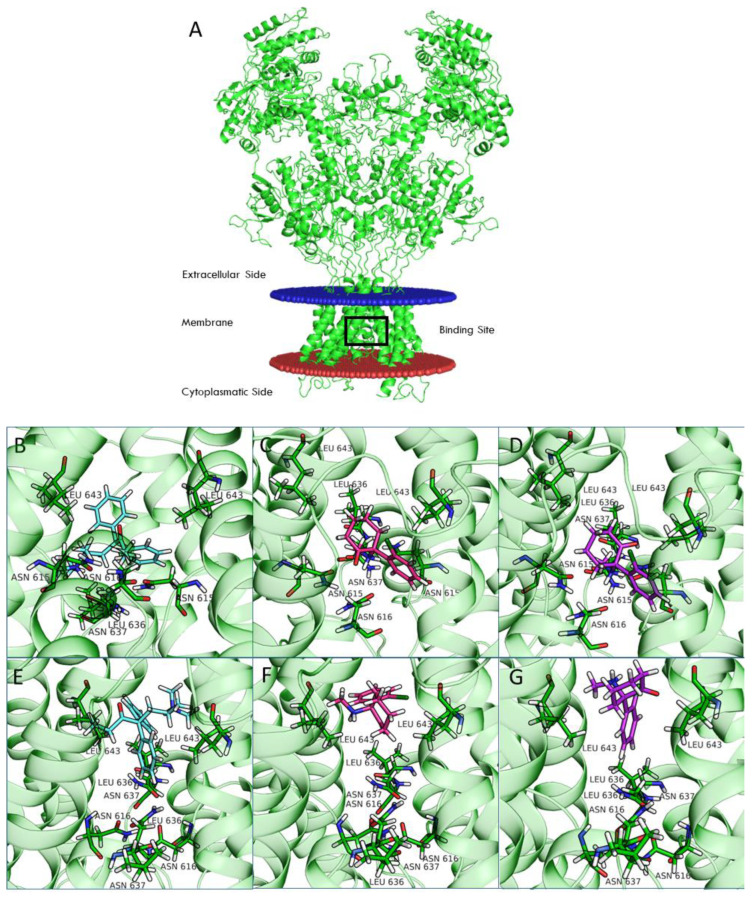
Structure of NR1-2D with drug binding site highlighted by a black box (**A**). The binding of esmethadone (**B**, light blue), arketamine (**C**, magenta), and esketamine (**D**, purple) in open (**B**–**D**, PDB code 6WHT) and closed (**E**–**G**, PDB code 6WHS) conformational states of NR1-2D are shown.

**Table 1 pharmaceuticals-15-00997-t001:** IC_50_ values of REL-1017 and reference NMDAR blockers.

	NR1-2A			NR1-2B			NR1-2C			NR1-2D		
	IC_50_ (µM)	Slope	Min. (%)	IC_50_ (µM)	Slope	Min. (%)	IC_50_ (µM)	Slope	Min. (%)	IC_50_ (µM)	Slope	Min. (%)
REL-1017	43	−1.0	30	25	−1.1	14	23	−0.84	15	68	−0.68	47
(±)-Ketamine	30	−0.76	23	6.3	−0.78	8	3.4	−0.83	8	11	−1.1	12
Memantine	34	−0.82	29	10	−0.86	11	3.6	−0.82	13	7.3	−0.88	18
Dextromethorphan	51	−0.80	35	15	−0.89	14	5.2	−1.0	14	28	−1.2	43
MK-801	0.29	−0.69	4	0.07	−0.94	4	0.58	−1.0	7	0.76	−1.2	11

IC_50_ values of five selected NMDAR channel blockers were obtained via FLIPR assay, as exemplified for REL-1017 in Figure 1. Fitting values were obtained for every heterodimeric NMDAR via logistic equation in GraphPad Prism v8.0. Slope is also reported in the table, as well as the minimal % Ca^2+^ influx measured in the presence of 100 μM blocker, the highest tested concentration. For example, 100 μM REL-1017 reduced Ca^2+^ influx elicited by 10 μM L-glutamate by 15% in the 2C-containing cell line.

**Table 2 pharmaceuticals-15-00997-t002:** K_B_ and affinity ratio values of REL-1017 and reference NMDAR blockers.

	NR1-2A		NR1-2B		NR1-2C		NR1-2D	
	K_B_ (µM)	Affinity Ratio (%)	K_B_ (µM)	Affinity Ratio (%)	K_B_ (µM)	Affinity Ratio (%)	K_B_ (µM)	Affinity Ratio (%)
REL-1017	8.9	51	6.1	74	4.5	100	7.8	58
(±)-Ketamine	4.3	11	1.1	42	0.46	100	1.4	33
Memantine	3.6	8	0.58	48	0.28	100	0.59	47
Dextromethorphan	9.6	13	1.9	63	1.2	100	6.7	18
MK-801	0.11	44	0.048	100	0.14	34	0.15	32

**Table 3 pharmaceuticals-15-00997-t003:** IC_50_ values of REL-1017 in presence of Mg^2+^.

	REL-1017 IC_50_ in 1 mM MgCl_2_	Hill Slope	Cell Number
NR1-2A	63.1	1.06	2-8
NR1-2B	41.7	1.17	2-7
NR1-2C	28.4	1.49	2-8
NR1-2D	13.5	1.42	3-7

Experiments were conducted in whole-cell patch-clamp electrophysiology at a holding potential of −60 mV. REL-1017 CRCs were obtained via whole-cell manual patch-clamp recordings in the presence of sub-saturating 1 μM L-glutamate, 10 μM glycine, and 1 mM MgCl_2_. Every clamped cell was assessed with a single concentration of REL-1017, and the cell number range indicates the minimum and the maximum number of clamped cells per concentration for each NMDAR subunit-expressing cell type. REL-1017 was found to be approximately five-fold more potent in blocking NR1-2D subtypes compared to NR1-2A subtypes. Fittings parameters for REL-1017 were obtained from data shown in Appendix A and analyzed with GraphPad Prism v8.0.

**Table 4 pharmaceuticals-15-00997-t004:** Onset kinetic parameters.

	Tau on (s)	I_1_ (% Current)	Cell Number
10 µM REL-1017	4.7 ± 0.21	20.4%	11
1 µM (±)-ketamine	4.7 ± 0.14	28.7%	10
10 µM (±)-ketamine	0.99 ± 0.050 ****	3.6%	4

Experiments were conducted in whole-cell patch-clamp electrophysiology at a holding potential of −70 mV. Onset kinetic constant (tau on) of an NMDAR antagonist represents the time required for the test item to reach approximately 63.2% of its current blocking effect. Tau on (s) is reported as mean ± SEM, while I_1_ is the % current measured at the end of test item addition and normalized with respect to I_0_ current, the current value before test item addition, which is set equal to 100%. Data passing normality (Shapiro–Wilk) test were analyzed with one-way ANOVA followed by Tukey’s test. **** is *p* < 0.0001. s is seconds.

**Table 5 pharmaceuticals-15-00997-t005:** Offset kinetic parameters.

	Tau off (s)	I_2_ (% Current)	Cell Number
10 µM REL-1017	17.7 ± 1.0	98.5%	11
1 µM (±)-ketamine	15.2 ± 0.63	95.5%	10
10 µM (±)-ketamine	17.2 ± 3.0	102.9%	4

Experiments were conducted in whole-cell patch-clamp electrophysiology at a holding potential of −70 mV. Offset kinetic constant (tau off) of an NMDAR antagonist represents the time required for the removal of approximately 63.2% of the current blocking effect of the test molecules, in continuous perfusion with a buffer containing the agonist L-glutamate but not the test item. Tau off (s) is reported as mean ± SEM, while I_2_ (% current) is measured at the end of test item removal and normalized with respect to I_0_ current, the current value before test item addition, which is set equal to 100%. Data passing normality (Shapiro–Wilk) test were analyzed with one-way ANOVA followed by Tukey’s test. Tau off values results were not significantly different. s is seconds.

**Table 6 pharmaceuticals-15-00997-t006:** Trapping parameters.

	B	B_R_	B_T_	Cell Number
10 µM REL-1017	83.8 ± 1.2 ****	71.8 ± 1.1 ***	85.9 ± 1.9	13
1 µM (±)-ketamine	74.0 ± 1.2	64.1 ± 1.3	86.7 ± 1.8	11

Experiments were conducted in whole-cell patch-clamp electrophysiology at a holding potential of −70 mV. Trapped block (B_T_) results were similar for 10 µM REL-1017 and 1 µM (±)-ketamine, although their initial block (B) and residual block (B_R_) were significantly different. Experiments were carried out as exemplified in Figure 5. Table data are mean ± SEM. Data passing normality (Shapiro–Wilk) test were analyzed with two-tailed unpaired T test. **** is *p* < 0.0001; *** is *p* < 0.001.

**Table 7 pharmaceuticals-15-00997-t007:** Glide docking score.

Ligands	NR1-2A		NR1-2B		NR1-2C		NR1-2D	
	Template 6WHS	Template 6WHT	Template 6WHS	Template 6WHT	Template 6WHS	Template 6WHT	Template 6WHS	Template 6WHT
REL-1017	−6.5	−8.0	−6.7	−8.0	−6.7	−8.1	−6.4	−8.1
MK-801	−7.0	−7.0	−7.2	−6.7	−7.1	−6.7	−7.1	−6.7
Arketamine	−6.1	−6.6	−7.0	−6.6	−6.2	−6.6	−6.3	−6.6
Esketamine	−6.1	−6.5	−6.4	−6.5	−6.0	−6.0	−6.0	−6.0

The Glide docking score (kcal/mol) of some of the investigated ligands at the NMDAR model of closed (6WHS) and open (6WHT) channel conformation.

## Data Availability

Data is contained within the article and supplementary material.

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
