# Peer review of "Pharmacological Comparative Characterization of REL-1017 (Esmethadone-HCl) and Other NMDAR Channel Blockers in Human Heterodimeric N-Methyl-D-Aspartate Receptors"

_pharmaceuticals, 2022, doi:10.3390/ph15080997_

Round 1

Reviewer 1 Report

This manuscript by Bettini et al., comprehensively compared the characteristics of  several uncompetitive NMDAR channel blockers, especially focused on REL-1017 (esmethadone-HCl), a novel uncompetitive NMDAR channel blocker in Phase 3 trials for the treatment of major depressive disorder . They demonstrated that the pharmacological characteristics of REL-1017 at NMDARs, including relatively low affinity at the NMDAR, NR1-2D subtype preference, the tau off and degree of trapping are similar to (±)- ketamine, and preferential docking and undocking of the open NMDAR. The experiments are well executed, the data are convincing and the statistical analysis is appropriate.

1, the uncompetitive NMDAR channel blockers seemed to block the tonic Ca2+ currents -spontaneous NMDAR-mediated miniature excitatory postsynaptic currents (NMDAR-mEPSCs). It is better to show whether REL-1017 block or depress NMDAR-mEPSCs in cortical brain slice?

2, In general the discussion seemed too lengthy and repeated description. The results and the discussion section of the manuscript could benefit from a revision into more concise writing style

Author Response

Reviewer #1

This manuscript by Bettini et al., comprehensively compared the characteristics of several uncompetitive NMDAR channel blockers, especially focused on REL-1017 (esmethadone-HCl), a novel uncompetitive NMDAR channel blocker in Phase 3 trials for the treatment of major depressive disorder. They demonstrated that the pharmacological characteristics of REL-1017 at NMDARs, including relatively low affinity at the NMDAR, NR1-2D subtype preference, the tau off and degree of trapping are similar to (±)- ketamine, and preferential docking and undocking of the open NMDAR. The experiments are well executed, the data are convincing, and the statistical analysis is appropriate.

1, the uncompetitive NMDAR channel blockers seemed to block the tonic Ca2+ currents -spontaneous NMDAR-mediated miniature excitatory postsynaptic currents (NMDAR-mEPSCs). It is better to show whether REL-1017 block or depress NMDAR-mEPSCs in cortical brain slice?

Response: We thank the Reviewer for highlighting the comprehensive nature of this work. As the reviewer indicates, future studies of REL-1017 blocking effects on NMDAR-mEPSCs in cortical brain slices may add informative data to these results. This point has been added to the discussion (see page 18, lines 602-603 of the revised manuscript):

Future studies of REL-1017 blocking effects on NMDAR-mEPSCs in cortical brain slices may add informative data to these results.

2, In general the discussion seemed too lengthy and repeated description. The results and the discussion section of the manuscript could benefit from a revision into more concise writing style

Response: The discussion was revised by deleting the last paragraph (page 17 lines 598-601) that was repetitive.

Reviewer 2 Report

This manuscript by Bettini et al. evaluates the effect of REL-1017, a novel non-competitive NMDA receptor antagonist, on various heterodimeric NMDA receptors (containing NR1-2A through 2D subunits) over-expressed in CHO cells, in both presence and absence of external magnesium. Experiments in absence of magnesium are carried out using a fluorometric imaging plate reader. Experiments in the presence of magnesium are carried out with either automated or manual whole-cell patch clamp. Additional whole-cell work measures onset and offset kinetics and evaluates susceptibility of tested compounds to ‘trapping’ in a deep NMDAR binding site. Finally, radioligand binding data (using cerebral cortex of Wistar rats) and in silico modeling of the receptor-antagonist interaction are also presented. At nearly every stage, data from REL-1017 is carefully compared to that obtained with other non-competitive NMDAR antagonists with known/varying effectiveness for treating resistant major depressive disorder and for inducing psychomimetic effects in humans. These compounds used for comparison include memantine, MK-801, dextromethorphan, and importantly, ketamine.

In brief, results indicate REL-1017 has slightly lower IC50 for NR1-2B and NR1-2C in absence of extracellular magnesium, and yet preferentially blocks NR1-2D in presence of extracellular magnesium. In each case, REL-1017 had higher IC50 than ketamine. Whole cell experiments indicate 10 uM REL-1017 and 1 uM ketamine both inhibit the response to 10 uM glutamate (as reported by NR1-2C containing NMDARs) by about 75%. Interestingly, by contrast, kinetic studies and trapping protocols highlight that REL-1017 and ketamine have similar susceptibility to trapping (again in assay using NR1-2C containing receptors). Radioligand binding studies confirmed higher IC50 and Ki values for REL-1017 than ketamine and other tested compounds. Computational modeling studies suggest REL-1017 has a higher preference for binding the open-channel conformation than esketamine, and may bind not quite as deep in the pore (perhaps owing in part to its larger size).

Overall, this is an extensive and outstanding study that I expect to be of significant interest in the scientific community. Core strengths of the manuscript include the extensive amount of high-quality experimental work, the range of technical approaches applied, the careful consideration of the effects of REL-1017 not only across a range of NMDAR subtypes, but also against both saturating and non-saturating concentrations of glutamate. Substantial additional strength comes from the extensive comparison of results to those obtained with ketamine and other non-competitive NMDAR antagonists. In my opinion, the work has very high potential impact not only for what it reveals about REL-1017 in particular, but probably even more so for the insight it provides on how affinity and trapping (and even charge) of any noncompetitive NMDAR antagonist may be related to therapeutic and psychotomimetic effects of these compounds. The discussion on this topic is well-written and extremely interesting (ultimately suggesting the combination of high trapping and low affinity in any non-competitive NMDAR antagonist may most effectively support therapeutic effects while avoiding dissociative effects, respectively). In comparison, weaknesses are minor.

My only somewhat significant comment for the authors is that I was confused about why NR1-2C was chosen for the kinetic studies in section 2.3.  The text on lines 259-260 says it is because all ‘clinically tolerated NMDAR blockers’ showed the lowest IC50 and Kb values against NR1-2C (in CHO cells expressing recombinant receptors). It’s unclear which compounds are meant exactly by ‘clinically tolerated’ here, and in any event, I’d have preferred to see these experiments run against NR1-2D because REL-1017 has lowest IC50 against that in presence of magnesium as presented in Table 3 (and because of the general interest in NR1-2D as a subunit that may be / likely is prominent in many extrasynaptic NMDARs in the CNS). As such, some further comment from the authors on the choice of NR1-2C for these experiments, and/or speculation on possible applicability to NR1-2D would be appreciated.

Other comments/suggestions are very minor:

 -the amended Figure S2 is the only figure in the manuscript that uses colors in any kind of line plot, but use of colorblind safe colors could probably be of benefit in a number of other figures as well (esp. those like Figs 1 and 2 with multiple overlapping line plots).

- %Effect as a y-axis label (e.g. as in figures 1, 2, and S1) is not particularly informative. Something like RFU (%max) would probably be better (where RFU = relative fluorescence units).

- If necessary data is available, a supplemental figure showing CRC for REL-1017 vs. ketamine (and perhaps other compounds tested) at least at NR1-2D might be valuable. If all data is available, plots like Fig. 1, but showing all compounds against each subunit (one plot / panel per subunit) would be a nice supplement to table 1 that the authors could consider.

- Need more consistency in use of REL-1017 vs. ‘esmethadone’ in several places, esp. in Fig. 4. Suggest sticking with REL-1017 throughout.

-Need more consistency in font sizes across figures. Again F4 is a clear example.

Author Response

­­­­­­­­Reviewer #2

This manuscript by Bettini et al. evaluates the effect of REL-1017, a novel non-competitive NMDA receptor antagonist, on various heterodimeric NMDA receptors (containing NR1-2A through 2D subunits) over-expressed in CHO cells, in both presence and absence of external magnesium. Experiments in absence of magnesium are carried out using a fluorometric imaging plate reader. Experiments in the presence of magnesium are carried out with either automated or manual whole-cell patch clamp. Additional whole-cell work measures onset and offset kinetics and evaluates susceptibility of tested compounds to ‘trapping’ in a deep NMDAR binding site. Finally, radioligand binding data (using cerebral cortex of Wistar rats) and in silico modeling of the receptor-antagonist interaction are also presented. At nearly every stage, data from REL-1017 is carefully compared to that obtained with other non-competitive NMDAR antagonists with known/varying effectiveness for treating resistant major depressive disorder and for inducing psychomimetic effects in humans. These compounds used for comparison include memantine, MK-801, dextromethorphan, and importantly, ketamine.

In brief, results indicate REL-1017 has slightly lower IC50 for NR1-2B and NR1-2C in absence of extracellular magnesium, and yet preferentially blocks NR1-2D in presence of extracellular magnesium. In each case, REL-1017 had higher IC50 than ketamine. Whole cell experiments indicate 10 uM REL-1017 and 1 uM ketamine both inhibit the response to 10 uM glutamate (as reported by NR1-2C containing NMDARs) by about 75%. Interestingly, by contrast, kinetic studies and trapping protocols highlight that REL-1017 and ketamine have similar susceptibility to trapping (again in assay using NR1-2C containing receptors). Radioligand binding studies confirmed higher IC50 and Ki values for REL-1017 than ketamine and other tested compounds. Computational modeling studies suggest REL-1017 has a higher preference for binding the open-channel conformation than esketamine and may bind not quite as deep in the pore (perhaps owing in part to its larger size).

Overall, this is an extensive and outstanding study that I expect to be of significant interest in the scientific community. Core strengths of the manuscript include the extensive amount of high-quality experimental work, the range of technical approaches applied, the careful consideration of the effects of REL-1017 not only across a range of NMDAR subtypes, but also against both saturating and non-saturating concentrations of glutamate. Substantial additional strength comes from the extensive comparison of results to those obtained with ketamine and other non-competitive NMDAR antagonists. In my opinion, the work has very high potential impact not only for what it reveals about REL-1017 in particular, but probably even more so for the insight it provides on how affinity and trapping (and even charge) of any noncompetitive NMDAR antagonist may be related to therapeutic and psychotomimetic effects of these compounds. The discussion on this topic is well-written and extremely interesting (ultimately suggesting the combination of high trapping and low affinity in any non-competitive NMDAR antagonist may most effectively support therapeutic effects while avoiding dissociative effects, respectively). In comparison, weaknesses are minor.

My only somewhat significant comment for the authors is that I was confused about why NR1-2C was chosen for the kinetic studies in section 2.3.  The text on lines 259-260 says it is because all ‘clinically tolerated NMDAR blockers’ showed the lowest IC50 and Kb values against NR1-2C (in CHO cells expressing recombinant receptors). It’s unclear which compounds are meant exactly by ‘clinically tolerated’ here, and in any event, I’d have preferred to see these experiments run against NR1-2D because REL-1017 has lowest IC50 against that in presence of magnesium as presented in Table 3 (and because of the general interest in NR1-2D as a subunit that may be / likely is prominent in many extrasynaptic NMDARs in the CNS). As such, some further comment from the authors on the choice of NR1-2C for these experiments, and/or speculation on possible applicability to NR1-2D would be appreciated.

Response:

We agree with Reviewer 2 that lines 259-260 need clarification and that the same experiment in NR1-2D subtypes may add informative data.

We changed the sentence (pages 8-9, lines 265-270) from:

”Because all clinically tolerated NMDAR blockers showed the lowest IC50 and KB values in recombinant cells expressing the NR1-2C subtype in the FLIPR assay, we selected the NR1-2C subtype for this study.”

To:

“With the exception of MK-801, the tested NMDAR blockers in clinical use (ketamine, memantine and dextromethorphan) or with potential for clinical uses (REL-1017) all showed the lowest IC50 and KB values in recombinant cells expressing the NR1-2C subtype in the FLIPR assay. Therefore, we selected the NR1-2C subtype for this study. Future trapping studies with the NR1-2D subtype may add to the NR1-2D data.”

Other comments/suggestions are very minor:

 -the amended Figure S2 is the only figure in the manuscript that uses colors in any kind of line plot, but use of colorblind safe colors could probably be of benefit in a number of other figures as well (esp. those like Figs 1 and 2 with multiple overlapping line plots).

Response: We have revised Figures 1 and 2 and used colorblind safe colors.

- %Effect as a y-axis label (e.g. as in figures 1, 2, and S1) is not particularly informative. Something like RFU (%max) would probably be better (where RFU = relative fluorescence units).

Response: As suggested by the Reviewer, we have corrected the y-axis label of Figures 1, 2, and S1, which is now “RFU (% max)”

- If necessary data is available, a supplemental figure showing CRC for REL-1017 vs. ketamine (and perhaps other compounds tested) at least at NR1-2D might be valuable. If all data is available, plots like Fig. 1, but showing all compounds against each subunit (one plot / panel per subunit) would be a nice supplement to table 1 that the authors could consider.

Response: As a supplement to table 1, we have added a new supplementary Figure (Figure S2) showing the effects of all compounds against each receptor subtype. 

- Need more consistency in use of REL-1017 vs. ‘esmethadone’ in several places, esp. in Fig. 4. Suggest sticking with REL-1017 throughout.

-Need more consistency in font sizes across figures. Again F4 is a clear example.

Response: We have changed esmethadone to REL-1017, and corrected font sizes across figures throughout the figures.

A missing comma was added at line 418.

A missing statement “and Figure S2” was added at line 139

We corrected from (Figure S6) to (Figure S7) at line 220

We corrected from Figure S7) to Figure S8) at line 239

We corrected from S3 and S4 to S4 and S5 at line 788